# Monetary Valuation of Children’s Cognitive Outcomes in Economic Evaluations from a Societal Perspective: A Review

**DOI:** 10.3390/children8050352

**Published:** 2021-04-29

**Authors:** Scott D. Grosse, Ying Zhou

**Affiliations:** National Center on Birth Defects and Developmental Disabilities, CDC, 4770 Buford Highway, Atlanta, GA 30341, USA; ips1@cdc.gov

**Keywords:** economic evaluation, cost-of-illness, cost-effectiveness, IQ, productivity costs, newborn screening, immunization, environmental health

## Abstract

Cognitive ability in childhood is positively associated with economic productivity in adulthood. Expected gains in economic output from interventions that protect cognitive function can be incorporated in benefit–cost and cost-effectiveness analyses conducted from a societal perspective. This review summarizes estimates from high-income countries of the association of general cognitive ability, standardized as intelligence quotient (IQ), with annual and lifetime earnings among adults. Estimates of the association of adult earnings with cognitive ability assessed in childhood or adolescence vary from 0.5% to 2.5% per IQ point. That range reflects differences in data sources and analytic methods. We take a conservative published estimate of a 1.4% difference in market productivity per IQ point in the United States from a recent study that controlled for confounding by family background and behavioral attributes. Using that estimate and the present value of lifetime earnings calculated using a 3% discount rate, the implied lifetime monetary valuation of an IQ point in the United States is USD 10,600–13,100. Despite uncertainty and the exclusion of non-market productivity, incorporation of such estimates could lead to a fuller assessment of the benefits of public health and clinical interventions that protect the developing brains of fetuses, infants, and young children.

## 1. Introduction

Experts increasingly recommend that economic evaluations of prevention strategies be conducted from either a societal perspective, which includes impacts on other sectors of the economy, or a limited societal perspective that encompasses impacts on time costs and economic productivity [1]. Inclusion of productivity gains is of particular salience for the assessments of strategies that target early childhood, because optimal health and development can raise employment and earning potential over the lifespan. For example, long-term outcome studies have demonstrated that disadvantaged children who receive intensive early interventions can experience lasting gains in educational attainment and occupational success as well as other benefits, the monetary valuation of which can greatly exceed the costs of the interventions [2,3]. Methods for the calculation of economic benefits of improved neurodevelopment may be useful for a range of interventions.

Adverse impacts on children’s neurodevelopment can be mitigated through prevention or the early detection and treatment of exposures that cause cognitive impairment. In general, cognitive impairment is defined as limitations in cognitive ability. Intellectual disability (ID) is a chronic condition defined as significant limitations in both cognitive ability and adaptive behavior that are recognized before the age of 22. Most clinicians measure cognitive ability with intelligence quotient (IQ) tests and define cognitive limitations as at least one standard deviation (SD) below the mean, and significant cognitive limitations as at least two SDs below the mean (e.g., roughly 70 or less on a scale in which the mean is 100 and the standard deviation is 15) [4,5]. Some of the most common causes of ID in the general population are genetic syndromes (e.g., Down syndrome and fragile X syndrome), fetal alcohol spectrum disorder, and adverse birth outcomes (e.g., premature birth) [5]. Sociodemographic variables, e.g., low maternal education and poverty, are associated with mild ID in the general population [6,7].

Economic evaluations of interventions to protect cognitive ability in young children often include estimates of the avoided costs of ID. For example, newborn screening followed by the prompt diagnosis and treatment for conditions such as phenylketonuria (PKU), congenital hypothyroidism (CH), and medium-chain acyl-CoA dehydrogenase deficiency can prevent ID and avert associated developmental consequences and economic costs [8,9,10,11]. Economic analysis of newborn screening for CH projected at least USD 2 in averted education and productivity costs associated with ID per USD 1 spent on screening [12]. Similarly, economic evaluations of vaccination against certain infections, notably bacterial meningitis and congenital cytomegalovirus, include estimates of avoided productivity and direct costs and lost productivity for cases of ID prevented [13,14,15].

The reported prevalence of ID is about 1.2% in U.S. children [16]. Many more persons can have milder degrees of cognitive impairment, although few economic assessments have incorporated estimates of impacts on the whole spectrum of cognitive ability. The one area where such assessments are routinely conducted involves environmental exposures to heavy metals, such as lead, which has dose–response associations with general intelligence, the ability to pay attention, and educational attainment [17,18]. Economic assessments of reductions in environmental lead exposure have typically modeled economic gains from a shift in the distribution of IQ scores [19]. For example, Grosse et al. calculated that the decline in blood lead levels between the late 1970s and early 1990s may have resulted in an average increase of 2.2–4.7 IQ points for each birth cohort, and as a result, each year’s birth cohort of four million children would earn at least USD 110 billion more (in 2000 USD) over their lifespans using a 3% annual discount rate [20]. Other authors have modeled economic gains from improvements in children’s intelligence attributable to full breastfeeding [21] and fortification for the prevention of iron-deficiency anemia [22].

Economic evaluations that restrict estimates of lost productivity to avoided cases of ID may understate the economic gains associated with the protection of brain development in children overall. For example, while 15–20% of children with spina bifida experience mild to severe ID [23], the distribution of cognitive ability among children with spina bifida born with hydrocephalus appears to be shifted to the left by one SD (mean IQ of roughly 85) [24]. The implication is that most individuals born with spina bifida and hydrocephalus experience some reduction in cognitive potential, although most have IQ scores within the normal range. Economic evaluations of folic acid fortification policies that only include costs associated with physical and intellectual disability [25,26] may understate the economic benefits of the prevention of mild cognitive impairment associated with spina bifida and hydrocephaly.

Childhood immunizations can potentially lead to improvements in adult productivity, which could make certain vaccines more economically attractive [27,28,29]. Studies conducted in low- and middle-income countries have reported that the implementation of childhood immunizations was associated with significantly higher educational attainments and children’s cognitive scores [30,31,32]. An association of bacterial meningitis with IQ scores in particular has been demonstrated; adult British survivors of certain types of bacterial meningitis had mean IQ scores reduced by 4–5 points relative to siblings [33]. Nonetheless, although economic evaluations of vaccination have quantified the benefits of prevention of severe sequelae of bacterial meningitis [13,14], they have not incorporated overall gains in cognitive ability and adult productivity.

Likewise, cost-effectiveness analyses of newborn screening have consistently failed to take into account the fact that many late-treated children with PKU and CH achieve final IQ scores within the normal range, albeit lower than in the reference population [8,34]. For example, researchers in Sweden reported that among 39 children aged 7–9 years with clinically diagnosed CH born prior to the introduction of screening for CH, the mean IQ was 88, and range was 50–113; just three were educated in special schools for children with intellectual disability [35]. However, although cost-effectiveness analyses of newborn screening for such disorders have routinely included the avoided costs of managing ID, they have not taken into account productivity gains associated with higher IQ scores for children who would not have been diagnosed with ID in the absence of screening [9,10].

In this study, we critically review published estimates of monetary valuations of an IQ point, independent of ID, used in U.S. assessments of the economic burden of exposures or disorders and in economic evaluations of interventions or policies that improve cognitive ability in early life. Unless otherwise stated, all estimates are reported in 2016 USD adjusted using the Gross Domestic Product deflator [36].

## 2. Materials and Methods

### 2.1. Estimation of General Cognitive Ability

IQ tests measure general cognitive skills that comprise verbal and nonverbal components. IQ scores have been shown to fluctuate during children’s early development, although results from different intelligence tests generally agree with one another. From early adolescence through adulthood, IQ scores on average remain stable, although individual scores can vary [37]. In addition, individual performance on IQ tests can vary depending on personality traits, such as internal locus of control, motivation, and incentives [38,39], as well as acquired factual knowledge.

Given the limited availability of IQ test results, estimates of the association of cognitive ability with annual earnings typically use standardized achievement test scores as a proxy for IQ scores. In particular, many U.S. researchers use the Armed Forces Qualifying Test (AFQT) scores derived from the Armed Services Vocational Aptitude Battery (ASVAB). The AFQT is normalized like an IQ test with a mean of 100 and SD of 15. The AFQT measures the mastery of factual knowledge, such as trigonometry, unlike IQ tests, which are intended to measure fluid intelligence [40,41]. Moreover, although AFQT scores are correlated with IQ scores, AFQT scores can be strongly influenced by inter-individual differences in personality traits, such as anxiety and inattention, and are an imperfect proxy for general intelligence [38,42]. In addition, AFQT scores are highly correlated with family socioeconomic status [40,43]. Achievement test scores are a function of age and schooling attainment; therefore, economists have typically adjusted ASVAB or AFQT scores for age and years of schooling attained at the time tests were taken [41,44,45,46]. In addition, measurement error is an underappreciated source of potential bias in statistical estimates of the association of ability and earnings [47]. The AFQT scores are censored, providing relatively little discrimination among individuals with high ability, and the measurement error in the AFQT is negatively correlated with true ability, unlike with IQ scores [48].

### 2.2. Estimation of Percentage Increase in Earnings per IQ Point

Calculation of the USD valuation of an IQ point can be divided into three steps. The first step is the calculation of the expected percentage change in productivity associated with a 1% difference in general cognitive ability. The second step is the calculation of the present value of lifetime productivity. The third step involves the calculation of the monetary valuation of an IQ point through multiplication of the estimates from the first two steps.

The proportional increase in annual productivity with increased cognitive ability at various times over the life cycle is typically estimated through linear regression analyses of the natural logarithm of annual earnings that include standard predictors of earnings from labor economics along with a measure or proxy for general cognitive ability assessed in childhood or adolescence. The regression coefficient on the cognitive test score indicates the percentage difference in earnings for a one-point different in cognitive ability. That approach presumes a linear association of cognitive ability with the logarithm of earnings, for which there is mixed evidence [49,50].

The sources of published estimates summarized in the Results section are derived from previous reviews of the literature on estimates of the IQ–earnings association published up to 2014 [19,51,52,53], supplemented by searches of Google Scholar and PubMed to identify more recent studies.

### 2.3. Estimation of Lifetime Earnings

Lifetime earnings refers to the expected sum of earnings over the lifetime of a statistical individual who is representative of the general population. The lifetime is calculated using survival probabilities from population life tables, which are used to calculate life expectancy. Lifetime earnings are calculated by multiplying average annual earnings at each age by the probability of survival to that age, discounted to present values, and summed over all ages.

Economic evaluations typically calculate the net present value (NPV) of both long-term outcomes and costs by applying a “discount” rate to costs and outcomes in future years. The rationale for discounting is two-fold: a higher valuation of current health and income (i.e., time preference) and the opportunity cost of government borrowing [54,55]. Countries can specify standard discount rates, varying from 1.5% to 5%; since 1996, the standard discount rate in the United States has been 3% [54,56].

In the human capital approach to the assessment of productivity (indirect) costs, analysts calculate the expected economic output for a synthetic cohort of individuals of a given age over the remaining lifetime [57,58,59]. The loss to society resulting from the removal of a worker in this conceptual approach is the additional contribution they make to aggregate economic output, which can exceed money income received. Most U.S. analyses use gross employee compensation, inclusive of fringe benefits and other employer costs [58,60,61,62], although some use only reported wage and salary income [63].

It is also standard practice in U.S. human capital analyses to include estimates of the economic valuation of non-market services, e.g., household services. The valuation of household services produced by women was included in the first U.S. human capital estimates prepared by Weisbrod in 1961 in order to reduce the disparity in human capital valuations of premature deaths between women and men [57,59]. Since the 1990s, national estimates of average market productivity and the sum of market and non-market productivity have been published in various sources [58,60,61,62,64]. In those analyses, time spent producing non-market services was valued using a replacement cost approach, i.e., what it would cost to hire workers to perform household services [58,61,62]. Some analysts have used opportunity cost approaches (e.g., median wage) to assign a monetary valuation to unpaid time [65]. In studies conducted in other countries, human capital assessments are typically restricted to estimates of market productivity, implicitly placing no economic valuation on household or voluntary activities [63].

The human capital approach to the calculation of productivity costs (loss of output due to premature mortality and disability) is used in the vast majority (92–95%) of cost-of-illness analyses globally, although a few countries recommend analysts use the “friction cost” approach that only takes into account short-run productivity losses associated with the replacement of a worker [63]. In the United States, the Second Panel on Cost-Effectiveness in Health and Medicine in 2016 likewise recommended use of the human capital approach to estimate productivity costs in cost-effectiveness analyses (CEAs) that follow a societal or modified societal perspective [66]. The Second Panel also recommended that societal perspective CEAs subtract consumption during the added years of life from productivity to calculate net resource use [67]. However, because guidance on calculating consumption costs in childhood is lacking, CEAs of life-saving pediatric interventions that follow the Second Panel guidance may exclude productivity costs [68].

The human capital approach has been used to calculate the loss of productivity from disability that results in the partial or complete loss of earning potential. For example, Waitzman et al. calculated the percentages of surveyed adults with and without spina bifida who reported being unable to work or were limited in the amount or kind of work they could perform [25]. They assumed a 100% loss in economic productivity (both market and non-market work) for the difference in percentages unable to work and a 50% reduction in productivity for those limited in work due to spina bifida.

The calculation of the NPV of expected productivity for a synthetic cohort in the human capital approach follows three steps. First, analysts calculate expected constant-USD (corrected for inflation) annual earnings or economic production in future years on the basis of cross-sectional data for adults at various ages. Secondly, analysts adjust projected earnings in future years to account for projected gains in average labor productivity (output per worker). Thirdly, analysts apply an exponential discount rate to productivity in future years to calculate the NPV of expected lifetime earnings for a statistical individual in a given year and sum up the overall NPV.

Grosse, Krueger, and Pike recently published updated estimates of annual and lifetime market and non-market productivity for the U.S. population stratified by age and sex, based on 2016 data [62]. Like previous studies, they estimated market productivity by adjusting survey data on reported gross money earnings to include employer payments on payroll taxes and monetary benefits [58,60,61,64]. Unlike previous studies, which assumed future growth in productivity of 1.0% per year, they reported estimates assuming either 0.5% or 1.0% annual productivity growth. The lower bound of 0.5% annual productivity growth was included to reflect slower productivity growth experienced during 2000–2016 than was previously projected [62]. The authors estimated the monetary valuation of non-market productivity using data from the American Community Survey on hours of services generated within the household as well as volunteer work and multiplied this by the estimated cost of hiring tasks to be performed. Lifetime productivity was estimated twice, using discount rates of 3% and 7%, along with survival probabilities from the 2014 U.S. life tables.

### 2.4. Calculation of Present Value of an IQ Point

The multiplication of the NPV of lifetime earnings by the percentage difference in earnings associated with a one IQ point difference in cognitive ability is a straightforward calculation. However, that approach implicitly assumes that life expectancy is independent of childhood cognitive ability, which is not correct. Individuals with ID, primarily moderate to profound ID, have markedly lower life expectancy [69]. In addition, among the population without ID, higher cognitive test scores in early life are associated with lower mortality risk in middle and late adulthood, much of which appears to be mediated by differences in educational attainment and socioeconomic status [70,71]. Higher IQ may lead to higher lifetime income through both longer life expectancy and higher annual earnings; therefore, existing estimates that ignore the positive association with life expectancy presumably underestimate the overall association of IQ with lifetime earnings.

## 3. Results

### 3.1. Proportional Increase in Earnings with Increased Cognitive Ability

Economic assessments of child lead exposure prevention have frequently used estimates of the IQ–earnings association [20,72,73,74,75,76,77,78,79,80,81,82,83,84,85,86]. Analyses conducted in the 1980s for the U.S. Environmental Protection Agency (EPA) conservatively projected gains of 0.9–1.0% in earnings per one IQ point [72,87]. In a more comprehensive analysis published in 1994 that improved on the earlier EPA analyses, Schwartz projected a 1.76% increase in annual earnings per IQ point gain, of which 72% was mediated by education [72]. The following year, Salkever estimated that a one-point increase in cognitive ability would raise annual earnings by 2.09% for males and 3.63% for females, with a weighted average of 2.37% for both genders [73]. The indirect mediating effect of education accounted for 41% of the overall association of cognitive ability and earnings for males and 61% for females. Grosse et al. used a 2.0% base-case estimate of the linear coefficient of earnings per IQ point with a range from 1.76% to 2.37% [20].

Other estimates have been derived from regression analyses of the natural logarithm of earnings in which cognitive test scores are included as a predictor along with covariates. Notably, in 1995, Salkever reported empirical estimates of the association of test scores and annual earnings using data from the 1979 National Longitudinal Study of Youth (NLSY79) cohort of individuals who had been tested at ages 14–23 in 1980 with the ASVAB, which measures factual knowledge instead of general cognitive ability. Salkever used AFQT scores derived from the ASVAB to predict earnings in 1990 at ages 25 to 33 [73]. Salkever estimated direct effects of 1.24% for males and 1.40% for females of a one-point score difference on earnings, adjusting for years of education.

Other researchers have reported more modest associations between IQ and annual earnings in early adulthood. Like Salkever, we excluded estimates of the association of IQ with hourly earnings because much of the effect of cognitive ability on earned income is mediated through annual hours of paid work [88]. A Swedish cohort study found a roughly 20% difference in earnings from ages 20 to 65 associated with a 1 SD difference in IQ measured at ages 10 or 20 [89]. Unlike in U.S. data, the IQ–earnings association in the Swedish cohort was similar for males and females. One SD in ability is equivalent to 15 IQ points; thus, the Swedish study finding implies a roughly 1.3% difference in earnings per IQ point. A different Swedish study that used earnings data at younger ages found that one SD in cognitive ability, using a Swedish test equivalent to the AFQT, was associated with only a 10% difference in earnings [49]. A recent cohort study from Canada found smaller adjusted differences in earnings at ages 33–35 relative to a one SD difference in verbal IQ assessed at age five or six, by up to 7.4% for males and 10.4% for females [90]. The relatively weak association of IQ and earnings in the Canadian study might reflect the inclusion of statistical controls for several behavioral characteristics, notably inattention, which is correlated with IQ scores, and reliance on verbal IQ instead of a full-measure IQ score (Francis Vergunst, personal communication, 20 March 2020).

Zax and Rees used data from the Wisconsin Longitudinal Study of Social and Psychological Factors in Aspiration and Attainment (WLS), which followed a cohort of Wisconsin high school graduates from 1957 for four decades. In regressions of annual earnings for males in 1974 and 1992 on adolescent IQ scores without covariates, each one-point difference in IQ scores was associated with 0.75% higher annual earnings in 1974 (age 35) and 1.39% higher earnings in 1992 (age 53) [91]. In a regression analysis with controls for family background, one IQ point was associated with 0.58% higher earnings at age 35. However, these estimates may understate the association of cognitive ability with earnings in contemporary, representative U.S. samples. The WLS cohort excluded non-graduates and females and had very few nonwhite participants, i.e., groups that have higher returns to cognitive ability [88,92].

More recently, Lin, Lutter, and Ruhm used data for the NLSY79 cohort followed from age 20 to age 50 with the 2006 renormed AFQT scores and reported results of a regression model that included controls for family background and three behavioral attributes; a retrospective assessment of sociability at age six, and adult self-assessments of locus of control and self-esteem [92]. The results indicated that a 0.1 SD difference in cognitive ability (equivalent to 1.5 IQ points) was associated with earnings differences of 2.0% at age 30, 2.7% at age 40, and 3.3% at age 50 (equivalent to 1.3%, 1.8%, and 2.2% differences in earnings per one-point difference in ability, respectively). The authors estimated the effect of a 0.1 SD increase in cognitive ability on the NPV of lifetime labor income from age 20 to age 50 using a 3% discount rate of 2.09% overall, 1.67% for men and 2.66% for women. The fact that the lifetime 2.09% estimate was only slightly higher than the 2.0% estimate at age 30 reflects the lower association of cognitive ability and earnings for individuals in their early 20s as well as greater discounting of earnings at older ages. The 2.09% difference for 0.1 SD is equivalent to 1.4% difference per IQ point.

### 3.2. USD Estimates of Incremental Productivity per IQ Point

Multiple authors have estimated the USD valuation of IQ using relative differences in earnings with IQ in combination with standardized estimates of the NPV of lifetime market productivity. For example, Grosse et al. calculated the valuation of an IQ point as USD 14,500 (in 2000-value USD) in their base-case analysis, using an estimated USD 623,000 NPV of lifetime gross market earnings for the year 2000 [64] and a 2.0% difference in earnings per IQ point. Grosse et al. also reported a range from USD 12,700 to USD 17,200 corresponding to 1.76% and 2.37% relative difference per IQ point, respectively [20]. Two subsequent studies took the USD 14,500 base-case valuation from Grosse et al. and adjusted it for inflation [78,79]. Other studies either used the rough USD 18,000 estimate in 2006 USD from the second of those studies without inflation adjustment [81,82,83] or modified it for purchasing power [85].

A few authors have a reported substantially higher human capital estimates of the valuation of IQ despite using the same percentage differences in earnings used in previous studies. Trasande and Liu in 2011 reported a USD 26,000 estimate of the valuation of an IQ point in 2007 USD. They applied the 2.0% difference in earnings from Grosse et al. to an unpublished (and unexplained) USD 1.3 million estimate of lifetime productivity [80], citing unpublished tabulations of lifetime earnings from Wendy Max and colleagues. Attina and Trasande followed the same approach in 2013, using the 1.76–2.37% difference per IQ point range [84], and a recent publication by Boyle et al. borrowed a cost estimate of USD 26,553 per IQ point from that study [86]. Attina and Trasande reporting estimates of discounted lifetime earnings at males and females aged five of USD 1,413,313 and USD 1,156,157, respectively, in 2007 USD, citing a report by Max et al. which in fact reported lifetime earnings estimates for 1–4-year-old boys and girls of USD 1,085,807 and USD 803,102, respectively, in 2000 USD [60]. They may have used a tabulation of combined market and non-market production provided to them directly, but that was not documented or explained. Grosse, Krueger, and Mvundura reported USD 1.2 million as the NPV in early childhood of combined market and non-market (household) productivity in 2007 USD with a 3% discount rate; the NPV of market productivity alone was USD 821,000 [61].

Updated estimates for the U.S. population in 2016 projected an NPV at birth of market productivity of USD 934,583 assuming 1% annual growth in future real earnings, and USD 758,954 assuming 0.5% annual growth productivity, both using a 3% discount rate [62]. The first valuation when multiplied by 2.0%, as in the 2002 study by Grosse et al., yielded a valuation of USD 18,700 per IQ point in 2016 USD, and multiplied by 1.4%, as in the study by Lin et al., which equates to a valuation of USD 13,100 per IQ point. Assuming annual productivity growth of 0.5%, the valuation of an IQ point is USD 10,600 if one assumes 1.4% higher earnings per IQ point, and USD 7600 if one assumes 1.0% higher earnings per IQ point.

Lin et al. calculated a valuation of USD 7862 per IQ point by applying the 1.4% difference in earnings per IQ point to the NPV of historical lifetime earnings reported by the NLSY79 cohort, extrapolated to age 65 [92]. This is comparable to an estimate using the NPV of future lifetime earnings in 2016 in combination with a conservative assumption of a 1.0% difference in earnings per IQ point. The use of historical earnings data can substantially understate expected earnings for future cohorts because of the long-term positive trend in real gross productivity and earnings.

## 4. Discussion

In addition to estimates of the economic costs associated with ID, the monetary valuation of IQ can be incorporated in both the burden of disease assessments and economic evaluations of preventive measures to enhance cognitive ability from the societal perspective. To date, estimates of the monetary valuation of IQ losses in the general population, as opposed to reductions in the occurrence of ID, have been incorporated in assessments of the economic burden of environmental contamination with lead, mercury, and other neurotoxicants, as well as in benefit–cost analyses (BCAs) of associated environmental regulations in the United States [19], but not in BCAs of interventions, such as newborn screening [11]. BCA experts recommend monetary valuations of major health outcomes, notably premature death, using either “willingness to pay (WTP)” estimates of utility gains in health or “willingness to accept (WTA)” losses in health [93]. However, BCA guidance suggests that other forms of monetary valuation, notably the human capital approach, can be appropriately used for the monetary valuation of mild adverse health impacts [94], such as mild cognitive losses caused by environmental contaminants.

Estimates of WTP/WTA valuations are derived from either revealed preference studies, in which real-world behaviors and outcomes are assessed, or stated preference studies, in which valuations are elicited through contingent valuation surveys or discrete choice experiments. In the United States, WTA estimates of the “value of a statistical life (VSL)” are based on “hedonic wage” regressions that relate occupational earnings to the risk of on-the-job death across occupations, controlling for other job and individual attributes [95,96]. The estimated VSL is typically about USD 10 million, independent of age [96,97,98], whereas the NPV of lifetime economic productivity for the U.S. population in 2016 was roughly USD 1.5 million at birth and USD 2 million at age 30 [62].

Limited attempts to generate revealed or stated preference WTA/WTP valuations of children’s cognitive ability in the United States have yielded valuations of either a range of USD 1600–2800 [99,100] or a point estimate of USD 600 per IQ point [101], all expressed in 2016 USD. The last estimate came from contingent valuation surveys of U.S. adults for WTP to reduce PCB contamination, which can affect children’s neurological development. Von Stackelberg and Hammitt noted that many of the respondents did not believe that PCB could lower IQ, and the authors speculated that the respondents were either not thinking of the effect of child IQ on adult earnings or heavily discounted the future [101]. The other WTP estimates came from a re-analysis by Lutter of contingent valuation data on parental WTP for lead chelation therapy for children with lead exposure collected by Agee and Crocker [99,100]. Lutter’s valuation estimates were used in a BCA prepared by critics of the regulation of mercury emissions [102]. Lutter noted, however, that only a small number of parents chose chelation therapy and that the average WTP for parents who chose therapy was USD 10,000 per IQ point [100]. This is roughly equivalent to human capital estimates of the NPV of post-tax lifetime earnings per IQ point, assuming 1% annual productivity growth and a 3% discount rate, which suggests that the two approaches may yield comparable estimates. However, evidence indicates that chelation therapy may have no long-term effects on children’s blood lead concentrations [103]. Given the extremely limited preference-based evidence available, further research might illuminate the difference in valuation of IQ using WTP and human capital approaches, if researchers are able to establish a validated method for assessing parental WTP for IQ differences.

Human capital estimates of the USD valuation of IQ as a function of the presumed log-linear association of annual earnings and cognitive ability could be characterized as an estimate of WTP for cognitive ability from the perspective of an employer who is purchasing work effort. In addition, a fully specified hedonic wage equation includes characteristics of individual workers, such as ability [95]. Therefore, use of human capital estimates of the monetary valuation of IQ based on the revealed preferences of employers in combination with individual behaviors in terms of educational attainment and labor force attachment may be justifiable within the WTP/WTA framework of welfare maximization used in BCA.

Human capital estimates of the USD valuation of IQ constitute the only currently accepted means for incorporating economic estimates of mild cognitive deficits in economic assessments. In addition to pediatric environmental health concerns, this approach has received some attention in the field of nutrition; specifically, assessments of the economic impacts of the prevention of micronutrient deficiencies through supplementation and of non-human milk infant feeding [21,22,104].

To apply this approach, cost-effectiveness analysts would need to estimate the gain in IQ points attributable to an intervention without double-counting benefits from the prevention of diagnosed disability, which may require extrapolation from limited data. For example, a Swedish retrospective screening study of stored blood spots from 100,239 children aged five years old, who had been screened at birth during 1977–1978 for PKU and galactosemia, administered the Griffith developmental assessment and laboratory tests to 26 children whose stored blood spots were positive for elevated thyrotropin (indicative of CH); six were found to be euthyroid and 20 had permanent hypothyroidism [105]. While two children had developmental quotients (DQs) of 45 and 58, the other 18 with CH had an average DQ of 95 (range 76–120), which was 12 points lower than the mean DQ of 107 (range 88–128) for six euthyroid children [105]. Assuming similar differences in mean DQ and IQ scores, a difference of 12 points and a monetary valuation of USD 10,600–13,100 per IQ point would imply an economic gain of USD 127,000–157,000 per U.S. infant detected with permanent CH through newborn screening; if non-market productivity gains were assumed, the estimates would be substantially greater.

Published estimates of the proportionate increase in earnings with general cognitive ability assessed in children and adolescents vary from roughly 0.5% to 2.5% per IQ point. Factors affecting the magnitude of estimates include the study population, the methods used to measure or impute cognitive ability, the age span over which earnings are assessed, and the covariates included in statistical analyses to control for confounding. Estimates of the USD valuation of cognitive ability also depend on the measure of productivity used—wages and salaries alone, gross earnings inclusive of fringe benefits or market earnings plus non-market productivity, assumptions about increases in productivity in future years, and the choice of discount rate.

Researchers may prefer to use combined market and non-market lifetime productivity to generate estimates of the monetary valuation of childhood IQ. To the extent that cognitive ability is positively associated with the performance of non-market activities, such as parenting, the expected economic valuation of cognitive ability could be substantially greater than is implied by the use of estimates of lifetime earnings. Total lifetime productivity based on 2016 U.S. data is estimated to be 57% greater than that of market productivity alone [62]. The studies reviewed here based estimates of the monetary valuation of IQ on U.S. data on what were said to be earnings data, although some may have actually used combined market and non-market productivity estimates [80,84,86]. On the other hand, some researchers used a human capital approach to estimate total productivity costs associated with physical and cognitive deficits. For example, Waitzman et al. explicitly assumed that adults with spina bifida and cerebral palsy experienced the same relative reductions in market and non-market productivity [25].

Estimates of a USD 10,600–13,100 valuation of an IQ point derived from the findings of Lin et al. on relative differences in earnings applied to the NPV of lifetime market productivity in the United States in 2016 USD are low relative to some estimates in the environmental health economics literature. Differences in the assumed percentage change in earnings per IQ point in part reflect differences in adjustments for potential confounding. Notably, Salkever did not adjust for confounding by family characteristics; failure to adjust for confounding can result in upwardly biased estimates of the association of ability and earnings [92,106,107,108].

Estimates of the monetary valuation of IQ are subject to substantial uncertainty. The fact that educational attainment can be both a confounder and a mediating variable in the causal pathway from ability to earnings creates inherent uncertainty in estimates of the causal effect of general intelligence on lifetime earnings. Much of the association between IQ and earnings appears to be mediated by educational attainment [89,108,109]; therefore, a regression of the effect of IQ on earnings that includes years of schooling captures only part of the overall effect of IQ. On the other hand, because education and intelligence are symbiotic, it is difficult to assess the separate effects of each on individual earning potential [110]. Individuals with higher cognitive ability are more likely to invest in post-secondary education and other training opportunities, and the returns to post-secondary education are greater for individuals with higher IQ test scores [111].

Estimates of the average association of cognitive ability with earnings do not necessarily apply equally to all groups. Discrimination and structural disadvantages can adversely affect both education and earnings. Nonetheless, it has been reported that the association of cognitive ability with earnings is stronger within groups experiencing discrimination, such as women and disadvantaged minorities [88]. Relatively little is known about the extent to which returns to cognitive ability vary depending on where one lies in the distributions of IQ scores and earnings. Two analyses of Swedish data on military enlistees linked to subsequent earnings both suggest that the association with cognitive ability is relatively stable across the distribution [49,50]. Lundborg et al. used unconditional quantile regression stratified by income deciles and reported that proportional gains in income (mostly earnings) with cognitive ability were generally higher for those in the upper half of the income distribution but were highest for those in the bottom decile [49]. However, that study did not stratify subjects by cognitive test scores. In a descriptive analysis stratified by cognitive test score, Lindqvist et al. reported that the steepest slope with earnings was among those with the lowest scores, and the slopes were similar for all other groups [50].

Estimates of the IQ–earnings association are also subject to potential bias due to measurement error. Classic (random) measurement error in cognitive tests can attenuate (downwardly bias) estimates of the associations of cognitive ability with variables, such as earnings [47]. On the other hand, non-random measurement error, omitted variables, and reverse causality can lead to overestimation of the effects of ability [112]. Heckman, Stixrud, and Urzua used corrected test scores and a latent variable model to assess both types of bias in the association of cognitive and noncognitive ability with hourly wages. The authors found that the downward bias of measurement error was offset by the upward biases from reverse causality and endogeneity, with the overall association of wages with cognitive ability modestly lower after correcting for both types of errors [112].

Another important source of confounding is the correlation of noncognitive skills or traits with cognitive ability. Behavioral traits, such as inattention and hyperactivity, can adversely affect the performance of cognitive tasks [113]. Those same traits can have substantial effects on adult earnings [110,114,115]. Indeed, studies have reported that personality traits or “soft skills”, such as conscientiousness, perseverance, sociability, and curiosity, can be even more predictive of adult earnings than cognitive ability [38,112]. Therefore, analyses that do not control for noncognitive characteristics may overstate the effects of cognitive ability [92]. Furthermore, behavioral traits, such as attentiveness and impulse control, have been shown to influence cognitive test scores [116].

We have presented a wide range of estimates of the IQ–earnings association. We advise caution in applying these estimates. Motivated reasoning can lead economic evaluators to select either high or low estimates of the monetary valuation of IQ, depending on their policy preferences. The use of a relatively high estimate will increase the likelihood that a policy which reduces the occurrence of cognitive deficits would be considered economically justified. Conversely, lower estimates of the IQ–earnings association can make interventions less likely to yield net benefit. This is not a hypothetical concern, as suggested by the different monetary valuations that have been used by advocates and critics of environmental regulations.

A monetary valuation of an IQ point in the range of USD 10,600–13,100 (2016 USD) per IQ point in the United States, assuming that lifetime earnings increase by 1.4% per IQ point and using a 3% discount rate, can be considered a conservative human capital-based estimate for two reasons. Firstly, those estimates do not take into account the positive association of childhood IQ with survival among working-age adults. Secondly, those estimates exclude gains in non-market productivity associated with higher cognitive ability. If cognitive ability were associated with relative productivity in non-market activities comparable to market productivity as quantified by earnings, the USD valuation would be in the range of USD 16,800–20,400 per IQ point. Three publications may have inadvertently used estimates of combined market and non-market productivity [80,84,86], but to date no published study has explicitly made that modeling choice or sought to justify the inclusion of non-market productivity.

Researchers have often borrowed U.S. monetary estimates of IQ valuations to apply to other countries [82,84,85]. A few researchers have generated their own estimates based on national estimates of earnings. Monahan et al. took the mid-point estimate from Zax and Rees of a 1.0% gain in earnings per IQ point and applied it to an estimate of average wage income in the United Kingdom from a labor force survey adjusted to assume 1% annual growth in future real earnings; their conservative base-case estimate of the monetary valuation of IQ was GBP 3297 [104]. This is lower than our U.S. estimates owing to between-country differences in average earnings, the exclusion of fringe benefits, and the use of a lower-bound estimate of the association of IQ and earnings. Monahan et al. used a range of estimates of IQ monetary valuations in a societal perspective CEA of a proposal for routine iodine supplementation of pregnant women in the United Kingdom for the prevention of cognitive impairment in children, which they concluded would be potentially cost-saving [104]. The authors noted that their estimates were conservative and cited reports of positive associations of IQ with various health outcomes in adults.

## 5. Conclusions

Although there is substantial imprecision and uncertainty in the available estimates of the monetary valuation of cognitive ability, we suggest that analysts consider the inclusion of such estimates in assessments of the economic benefits of public health and clinical interventions that protect the developing brains of fetuses, infants, and young children.

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
