# Peer review of "Monetary Valuation of Children’s Cognitive Outcomes in Economic Evaluations from a Societal Perspective: A Review"

_children, 2021, doi:10.3390/children8050352_

Round 1

Reviewer 1 Report

Dear authors,

Thanks for a very nice, interesting, and relevant study! I just have one major remark: I am not fully happy with the term " implied lifetime monetary value of an IQ point in the United States" because "value" is an ethical concept and the term may be misunderstood as if individuals with lower IQ were of lower value. Not sure how best to change it - at least, a perspective should be included (already in the abstract for quick readers) and the topic should be addressed in the discussion section; maybe you also find another wording or other ways to address this point.

And, a minor point: The duplication of the phrase "The first step" in the first two paragraphs of section 2.2 sounds a bit irritating to me. I'd propose rephrasing one of them.

Author Response

Thank you for your kind comments.

We have replaced “value” with “monetary valuation” throughout, including in the title.

We eliminated the second occurrence of “The first step.” We deleted the text, “…first step is to estimate the….”

Reviewer 2 Report

Please see document attached

Author Response

Thank you for your kind comments.

We have revised the description of the Schwartz 1994 article, “Analyses conducted in the 1980s for the U.S. Environmental Protection Agency (EPA) conservatively projected gains of 0.9-1.0% in earnings per 1 IQ point 72, 87. In a more comprehensive analysis published in 1994 that improved on the earlier EPA analyses, Schwartz projected a 1.76% increase in annual earnings per IQ point gain, of which 72% was mediated by education 72.”

We restructured the first sentence in the Discussion section, “In addition to estimates of the economic costs associated with ID, monetary valuation of IQ can be incorporated in both burden of disease assessments and economic evaluations of preventive measures to enhance cognitive ability from the societal perspective.”

Reviewer 3 Report

This paper reviews estimates of expected gains in economic output from interventions that protect cognitive function in children.  These gains can be used in economic evaluation such as benefit-cost analysis and cost-effectiveness analysis.  An estimate of a 1.4% difference in market productivity per IQ point and the present value of lifetime earnings are combined to yield a gain of $10,600-$13,100 per IQ point.  This range of value is suggested as a measure of benefits for interventions that affect IQ in children.

The authors are careful to consider several potentially important issues.  They recognize that gains could take the form of increases in longevity as well as higher earnings.  They acknowledge that gains could take the form of increases in non-market productivity in addition to market productivity.  They allow that higher IQ might increase non-cognitive characteristics as well as cognitive.

Two related sections of the paper could be improved with a bit more care.  The discussion of preference-based valuation compared to earnings-based valuation is somewhat unsettling (lines 369-388).  WTA/WTP estimates are described as relatively low as if the earnings-based values are the “true” measures of benefits.  Identifying the authors of a BCA that uses WTA/WTP values as critics of regulation of mercury emissions hints at the same thing, the earning-based measure is better.  The effect of IQ on earnings is a market effect that reflects marginal valuations of workers and employers, but it begs the question of why that isn’t incorporated in parents’ preference-based valuations of their children’s health.  Helicopter parents would incorporate the effect.  Barring dysfunctional parents, parents would be considered key members of society whose values should count.  A safer point might be to say we have few preference-based studies, the differences are not well-understood, and explanation awaits further research.   

The second section is the discussion of the monetary value of an IQ point, $10,600-$13,100, as a lower bound.  It should note that it includes only the earnings-based measure (lines 476-486).  The reasons for the sizable difference between the earnings-based values and preference-based values are unclear, but sensitivity analysis should include both since WTA/WTP measures are more theoretically correct.  Regardless of the range of estimates, the case is strong that analysts should include the values of interventions that protect developing brains of fetuses, infants, and young children.  We are certain that the benefits are positive, not zero.

Author Response

Thank you for your thoughtful comments.

We have substantially rewritten the second paragraph in the section on WTP/WTA estimates to provide more specific information about the two sets of WTP published estimates and their limitations. That paragraph now reads,

Limited attempts to generate revealed or stated preference WTA/WTP valuations of children’s cognitive ability in the United States have yielded valuations of either a range of $1600-$2800 99, 100 or a point estimate of $600 per IQ point 101, all expressed in 2016 dollars. The last estimate came from contingent valuation surveys of U.S. adults for WTP to reduce PCB contamination, which can affect children’s neurological development. Von Stackelberg and Hammitt noted that many of the respondents didn’t believe that PCB could lower IQ, and the authors speculated that the respondents were either not thinking of the effect of child IQ on adult earnings or heavily discounted the future 101. The other WTP estimates came from a re-analysis by Lutter of contingent valuation data on parental WTP for lead chelation therapy for children with lead exposure collected by Agee and Crocker 99, 100. Lutter’s valuation estimates were used in a BCA prepared by critics of regulation of mercury emissions 102. Lutter noted, however, that only a small number of parents chose chelation therapy and that the average WTP for parents who chose therapy was $10,000 per IQ point. 100 That is roughly equivalent to human capital estimates of the NPV of post-tax lifetime earnings per IQ point assuming 1% productivity growth and a 3% discount rate. That suggests the two approaches may yield comparable estimates. However, evidence indicates that chelation therapy may have no long-term effects on children’s blood lead concentrations 103. Given the extremely limited preference-based evidence available, further research might illuminate the difference in valuation of IQ using WTP and human capital approaches, if researchers were able to establish a validated method for assessing parental WTP for IQ differences.   

We have now modified the description of the $10,600-$13,000 range as “conservative,” stating that the range “can be considered a conservative human capital-based estimate for two reasons. That acknowledges that it is conservative among the set of human capital-based estimates. We do not repeat here our discussion of the limitations of the extant WTP/WTA estimates and the desirability of better estimates.